# A noise-corrected Langevin algorithm and sampling by half-denoising

**Aapo Hyvärinen**  *aapo.hyvarinen@helsinki.fi*
*Department of Computer Science*
*University of Helsinki, Finland*

**Reviewed on OpenReview:** *https://openreview.net/forum?id=QGtXn5GtfK*

## Abstract

The Langevin algorithm is a classic method for sampling from a given pdf in a real space. In its basic version, it only requires knowledge of the gradient of the log-density, also called the score function. However, in deep learning, it is often easier to learn the so-called "noisy-data score function", i.e. the gradient of the log-density of noisy data, more precisely when Gaussian noise is added to the data. Such an estimate is biased and complicates the use of the Langevin method. Here, we propose a noise-corrected version of the Langevin algorithm, where the bias due to noisy data is removed, at least regarding first-order terms. Unlike diffusion models, our algorithm only needs to know the noisy-data score function for one single noise level. We further propose a simple special case which has an interesting intuitive interpretation of iteratively adding noise the data and then attempting to remove half of that noise.

## 1 Introduction

A typical approach to generative AI with data in $\mathbb{R}^k$ consists of estimating an energy-based (or score-based) model of the data and then applying a sampling method, typically MCMC. Among the different methods for learning the model of the data, score matching is one of the simplest and has achieved great success recently. In particular, denoising score matching (DSM) is widely used since it is computationally very compatible with neural networks, resulting in a simple variant of a denoising autoencoder (Vincent, 2011). The downside is that it does not estimate the model for the original data but for noisy data, which has Gaussian noise added to it. This is why it is difficult to use standard MCMC methods for sampling in such a case. Henceforth we call the score function for the noisy data the "noisy-data score function". We emphasize that the noisy-data score function does not refer to an imperfectly estimated score function which would be noisy due to, for example, a finite sample; it is the exact score function for the noisy data, since this is what DSM gives in the limit of infinite data.

A possible solution to this problem of bias is given by diffusion models (Croitoru et al., 2023; Yang et al., 2023), which are used in many of the state-of-the-art methods (Rombach et al., 2022). Diffusion models need precisely the score function of the noisy data; the downside is that they need it for many different noise levels. Another solution is to estimate the noisy-data score for different noise levels and then extrapolate (anneal) to zero noise (Song and Ermon, 2019). However, such learning necessitating many noise levels is computationally costly.

One perhaps surprising utility of using the noisy-data score function for sampling might be that it is a kind of regularized version of the true score function. In particular, any areas in the space with zero data will have a non-zero probability density when noise is added. This might lead to better mixing of any MCMC (Saremi and Hyvärinen, 2019; Song and Ermon, 2019), and even improve the estimation of the score function (Kingma and Cun, 2010). Hurault et al. (2025) analyze the error in sampling using DSM and Langevin in detail, considering the Gaussian distribution.

Here, we develop an MCMC algorithm which uses the noisy-data score function, and needs it for a single noise level only. This should have computational advantages compared to diffusion models, as well as leading to a simplified theory where the estimation and sampling parts are clearly distinct. In particular, we develop a *noise-corrected Langevin* algorithm, which works using the noisy-data score function, while providing samples from the original, noise-free distribution. Such a method can be seamlessly combined with denoising score matching. A special case of the algorithm has a simple intuitive intepretation as an iteration where Gaussian noise is first added to the data, and then, based on the well-known theory by Tweedie and Miyasawa, it is attempted to remove that noise by using the score function. However, only half of the denoising step is taken, leading to an algorithm based on "half-denoising".

## 2 Background

We first develop the idea of denoising score matching based on the Tweedie-Miyasawa theory, and discuss its application in the case of the Langevin algorithm.

### 2.1 Score functions and Tweedie-Miyasawa theory

We start with the well-known theory for denoising going back to Miyasawa (1961); Robbins et al. (1956); see Raphan and Simoncelli (2011); Efron (2011) for a modern treatment and generalizations. They considered the following denoising problem. Assume the observed data $\tilde{\mathbf{x}}$ is a sum of original data $\mathbf{x}$ and some noise $\mathbf{n}$:

$$\tilde{\mathbf{x}} = \mathbf{x} + \mathbf{n} \tag{1}$$

Now, the goal is to recover $\mathbf{x}$ from an observation of $\tilde{\mathbf{x}}$. The noise $\mathbf{n}$ is assumed Gaussian with covariance $\boldsymbol{\Sigma}_{\mathbf{n}}$.

An important role here is played by the gradient of the log-derivative of the probability density function (pdf) of $\tilde{\mathbf{x}}$. We denote this for any random vector $\mathbf{y} \in \mathbb{R}^k$ with pdf $p_{\mathbf{y}}$ as:

$$\boldsymbol{\Psi}_{\mathbf{y}}(\boldsymbol{\eta}) = \nabla_{\boldsymbol{\eta}} \log p_{\mathbf{y}}(\boldsymbol{\eta}) \tag{2}$$

which is here called the *score function*.

One central previous result is the following theorem, often called an "empirical Bayes" theorem. Robbins et al. (1956); Efron (2011) attribute it to personal communication from Maurice Tweedie, which is why it is often called "Tweedie's formula", while it was also independently published by Miyasawa (1961).

**Theorem 1 (Tweedie-Miyasawa)**

$$\mathbb{E}\{\mathbf{x}|\tilde{\mathbf{x}}\} = \tilde{\mathbf{x}} + \boldsymbol{\Sigma}_{\mathbf{n}} \boldsymbol{\Psi}_{\tilde{\mathbf{x}}}(\tilde{\mathbf{x}}) \tag{3}$$

For the sake of completeness, the proof is given in the Appendix. The Theorem provides an interesting solution to the denoising problem, or recovering the original $\mathbf{x}$, since it gives the conditional expectation which is known to minimize mean-squared error. In practice, the price to pay is that we need to estimate the score function $\boldsymbol{\Psi}_{\tilde{\mathbf{x}}}$.

### 2.2 (Denoising) score matching

Another background theory we need is (non-parametric) estimation of the score function. Consider the problem of estimating the score function $\boldsymbol{\Psi}_{\mathbf{x}}$ of some data set. Suppose in particular that we use a neural network to estimate $\boldsymbol{\Psi}_{\mathbf{x}}$. Any kind of maximum likelihood estimation is computationally very difficult, since it would necessitate the computation of the normalization constant (partition function).

Hyvärinen (2005) showed that the score function can be estimated by minimizing the expected squared distance between the model score function $\boldsymbol{\Psi}(\mathbf{x}; \boldsymbol{\theta})$ and the empirical data score function $\boldsymbol{\Psi}_{\mathbf{x}}$. He further showed, using integration by parts, that such a distance can be brought to a relatively easily computable

form. Such score matching completely avoids the computation of the normalization constant, while provably providing a estimator that is consistent. However, due to the existence of higher-order derivatives in the objective, using the original score matching objective by Hyvärinen (2005) is still rather difficult in the particular case of deep neural networks, where estimating any derivatives of order higher than one can be computationally demanding (Martens et al., 2012). Therefore, various improvements have been proposed, as reviewed by Song and Kingma (2021).

Denoising score matching (DSM) by Vincent (2011) provides a particularly attractive approach in the context of deep learning. Given original noise-free data $\mathbf{x}$, it learns the score function of a noisy $\tilde{\mathbf{x}}$ with noise *artificially* added as in (1). Interestingly, denoising score matching can be derived from Theorem 1. Since the conditional expectation is the minimizer of the mean squared error, we have the following corollary, proven in Appendix B:

**Corollary 1** *Assume we observe both $\mathbf{x}$ and $\tilde{\mathbf{x}}$, which follow (1) with Gaussian noise $\mathbf{n}$. The score function $\boldsymbol{\Psi}_{\tilde{\mathbf{x}}}$ of the noisy data is the solution of the following minimization problem:*[1]

$$\min_{\boldsymbol{\Psi}} \mathbb{E}_{\tilde{\mathbf{x}},\mathbf{x}}\{\|\mathbf{x} - (\tilde{\mathbf{x}} + \boldsymbol{\Sigma}_{\mathbf{n}}\boldsymbol{\Psi}(\tilde{\mathbf{x}}))\|^2\} \tag{4}$$

We emphasize that unlike in the original Tweedie-Miyasawa theory, it is here assumed we observe the original noise-free $\mathbf{x}$, and add the noise ourselves to create a self-supervised learning problem. Thus, we learn the noisy-data score function by training what is essentially a denoising autoencoder.

The advantage of DSM with respect to the original score matching is that there are less derivatives; in fact, the problem is turned into a basic least-squares regression. The disadvantage is that we only get the noisy-data score function $\boldsymbol{\Psi}_{\tilde{\mathbf{x}}}$. This is a biased estimate, since what we want in most cases is the original $\boldsymbol{\Psi}_{\mathbf{x}}$.

### 2.3 Langevin algorithm

The Langevin algorithm is perhaps the simplest MCMC method to work relatively well in $\mathbb{R}^k$. The basic scheme is as follows. Starting from a random point $\mathbf{x}_0$, compute the sequence:

$$\mathbf{x}_{t+1} = \mathbf{x}_t + \mu \boldsymbol{\Psi}_{\mathbf{x}}(\mathbf{x}_t) + \sqrt{2\mu}\,\boldsymbol{\nu}_t \tag{5}$$

where $\boldsymbol{\Psi}_{\mathbf{x}} = \nabla_{\mathbf{x}} \log p(\mathbf{x})$ is the score function of $\mathbf{x}$, $\mu$ is a step size, and $\boldsymbol{\nu}_t$ is Gaussian noise from $\mathcal{N}(\mathbf{0}, \mathbf{I})$. According to well-known theory, for an infinitesimal $\mu$ and in the limit of infinite $t$, $\mathbf{x}_t$ will be a sample from $p$. The assumption of an infinitesimal $\mu$ can be relaxed by applying a "Metropolis-adjustment", but that complicates the algorithm, and requires us to have access to the energy function as well, which may be the reason why such a correction is rarely used in deep learning.

If we were able to learn the score function of the original $\mathbf{x}$, we could do sampling using this Langevin algorithm. But DSM only gives us the score function of the noisy data. This contradiction inspires us to develop a variant of the Langevin algorithm that works with $\boldsymbol{\Psi}_{\tilde{\mathbf{x}}}$ instead of $\boldsymbol{\Psi}_{\mathbf{x}}$.

## 3 Noise-corrected Langevin algorithm

Now, we proceed to propose a new "noise-corrected" version of the Langevin algorithm. The idea of noise-correction assumes that we have used DSM to estimate the score function of noisy data. This means our estimate of the score function is strongly biased. While this bias is known, it is not easy to correct by conventional means, and a new algorithm seems necessary.

---

[1]To clarify: $\boldsymbol{\Psi}$ is here any function used as an argument for minimization. It is denoted in the same way as the score functions, because eventually at the optimum, it will be an estimate of a score function.

### 3.1 Proposed algorithm

We thus propose the following noise-corrected Langevin algorithm. Starting from a random point $\mathbf{x}_0$, compute the sequence:

$$\mathbf{x}_{t+1} = \tilde{\mathbf{x}}_t + \mu \mathbf{\Psi}_{\tilde{\mathbf{x}}}(\tilde{\mathbf{x}}_t) + \sqrt{2\mu - \sigma^2}\,\boldsymbol{\nu}_t \tag{6}$$

with

$$\tilde{\mathbf{x}}_t = \mathbf{x}_t + \sigma \mathbf{n}_t, \qquad\qquad \mathbf{n}_t \sim \mathcal{N}(\mathbf{0}, \mathbf{I}), \qquad\qquad \boldsymbol{\nu}_t \sim \mathcal{N}(\mathbf{0}, \mathbf{I}) \tag{7}$$

where $\mu$ is a step size parameter. The function $\mathbf{\Psi}_{\tilde{\mathbf{x}}}$ is the score function of the noisy data $\tilde{\mathbf{x}}$ created as in (1), with noise variance being $\mathbf{\Sigma}_{\mathbf{n}} = \sigma^2 \mathbf{I}$. (As in previous sections, we use here $\mathbf{x}$ without time index to denote the original observed data, and $\tilde{\mathbf{x}}$ to denote the observed data with noise added, while the same letters with time indices denote related quantities in the MCMC algorithm.)

The main point is that this algorithm only needs the noisy-data score function $\mathbf{\Psi}_{\tilde{\mathbf{x}}}$, which would typically be given by DSM. The weight $\sqrt{2\mu - \sigma^2}$ of the last term in the iteration (6) is accordingly modified from the original Langevin iteration (5) where it was $\sqrt{2\mu}$. Clearly, we need to assume the condition

$$\mu \geq \sigma^2/2 \tag{8}$$

to make sure this weight is well-defined. Another important modification is that the right-hand side of (6) is applied on a noisy version of $\mathbf{x}_t$, adding noise $\mathbf{n}_t$ to the $\mathbf{x}_t$ itself at every step.

### 3.2 Theoretical analysis

#### 3.2.1 Semi-heuristic argument

First, we provide a simple analysis of the behaviour of the algorithm.[2] Add $\sigma \mathbf{n}_{t+1}$ on both sides of Eq. (6). Since $\tilde{\mathbf{x}}_{t+1} = \mathbf{x}_{t+1} + \sigma \mathbf{n}_{t+1}$, this becomes

$$\tilde{\mathbf{x}}_{t+1} = \tilde{\mathbf{x}}_t + \mu \mathbf{\Psi}_{\tilde{\mathbf{x}}}(\tilde{\mathbf{x}}_t) + \sqrt{2\mu - \sigma^2}\,\boldsymbol{\nu}_t + \sigma \mathbf{n}_{t+1} \tag{9}$$

We can collect the noise terms on the RHS together by defining

$$\bar{\boldsymbol{\nu}}_t = \frac{1}{\sqrt{2\mu}}\left[\sqrt{2\mu - \sigma^2}\,\boldsymbol{\nu}_t + \sigma \mathbf{n}_{t+1}\right] \tag{10}$$

This new noise $\bar{\boldsymbol{\nu}}_t$ follows $\mathcal{N}(\mathbf{0}, \mathbf{I})$. Moreover, it is independent of $\tilde{\mathbf{x}}_t$ and i.i.d. over time. Thus, our algorithm implies the following iteration for $\tilde{\mathbf{x}}_t$

$$\tilde{\mathbf{x}}_{t+1} = \tilde{\mathbf{x}}_t + \mu \mathbf{\Psi}_{\tilde{\mathbf{x}}}(\tilde{\mathbf{x}}_t) + \sqrt{2\mu}\,\bar{\boldsymbol{\nu}}_t \tag{11}$$

which is exactly an ordinary Langevin iteration for $\tilde{\mathbf{x}}_t$.

We see that $\tilde{\mathbf{x}}_t$ converges to sample noisy data from $p_{\tilde{\mathbf{x}}}$, under the same conditions and reservations as an ordinary Langevin iteration. In other words, $p_{\tilde{\mathbf{x}}_t} \to p_{\tilde{\mathbf{x}}}$ in the approximative sense of the Langevin algorithm. Now, we can simply deconvolve both sides of this limit equation, and get $p_{\mathbf{x}_t} \to p_{\mathbf{x}}$, in a heuristic sense. This provides a semi-heuristic proof that the algorithm samples from $p_{\mathbf{x}}$.

Of course, one can always use an iteration such as in Eq. (11) to sample $\tilde{\mathbf{x}}$ from the noisy distribution, given the noisy-data score. But then there is no straightforward way of computing the $\mathbf{x}_t$ which is not an obvious function of $\tilde{\mathbf{x}}_t$. By construction, our algorithm does compute the noise-free samples $\mathbf{x}_t$ as well.

However, an alternative perspective to our algorithm is to notice that after running ordinary Langevin iterations with noisy-data score as in (11), all that is needed is a *single* iteration of the new iteration (6) to get a sample from the noise-corrected algorithm, and thus the noise-free distribution. In fact, our algorithm was analyzed from this viewpoint by Beyler and Bach (2025), based on an earlier preprint of our paper.

---

[2]I'm grateful to the Action Editor, Atsushi Nitanda, for proposing the starting point of this analysis.

### 3.2.2 Rigorous analysis

To more rigorously understand the behavior of the algorithm, we compare it with what we call the *Oracle Langevin.* This means the ordinary Langevin iteration where the true score function $\mathbf{\Psi_x}$ (i.e. of the noise-free data $\mathbf{x}$) is known and used. Basically, our goal in this paper can be formulated as emulating the Oracle Langevin iteration while only knowing the score $\mathbf{\Psi_{\tilde{x}}}$ of the noisy data.

Our main theoretical result is the following Theorem, proven in the Appendix:

**Theorem 2** *Denote the target pdf by $p_\mathbf{x}$ and its score function as $\mathbf{\Psi_x}$. Assume the following regularity conditions on the target distribution:*

*1. The score function $\mathbf{\Psi_x}$ has bounded norm, the bound being denoted as*

$$C_\Psi = \sup_{\boldsymbol{\alpha} \in \mathbb{R}^d} \|\mathbf{\Psi_x}(\boldsymbol{\alpha})\| \tag{12}$$

*2. $p_\mathbf{x}$ is at least twice differentiable; $p_\mathbf{x}$ and its first two derivatives are square-integrable.*

*We are given the noisy-data score function $\mathbf{\Psi_{\tilde{x}}}$ to be used in our algorithm where $\tilde{\mathbf{x}}$ follows (1). Denote its associated pdf as $p_{\tilde{\mathbf{x}}}$ and define the constant*

$$C_\epsilon = \max(C_\Psi^2 \|p_{\tilde{\mathbf{x}}} - p_\mathbf{x}\|_2, C_\Psi \|p_{\tilde{\mathbf{x}}}(\mathbf{\Psi_x} - \mathbf{\Psi_{\tilde{x}}})\|_2) \tag{13}$$

*Consider the iteration of the algorithm proposed in Eq. (6) in the space of characteristic functions and at the true distribution $\mathbf{x}_t \sim p_\mathbf{x}$. Fix any given value for $\boldsymbol{\xi}$, the argument of the characteristic function $\hat{p}_{\mathbf{x}_t}(\boldsymbol{\xi})$.*

*Then, one iteration of the proposed algorithm is equal to one iteration of the Oracle Langevin up to terms of order $O(\max(\mu^3 C_\Psi^3 \|\boldsymbol{\xi}\|^3, \mu^3 \|\boldsymbol{\xi}\|^2, \mu^2 C_\epsilon \|\boldsymbol{\xi}\|^2))$.*

Thus, we prove that our iteration removes the bias due to estimation of the score from the noise in the data $\tilde{\mathbf{x}}$, up to higher-order terms, and at the true distribution $p_\mathbf{x}$. The results could presumably be extended to the vicinity of $p_\mathbf{x}$ which would give some more higher-order terms in the approximation.

The regularity assumptions here are sufficient but not necessary. We note that Assumption 1), i.e. a bounded score, is typically valid for super-Gaussian distributions (i.e. with heavy tails), such as the Laplace distribution. The addition of Gaussian noise implicit in $\mathbf{\Psi_{\tilde{x}}}$ has little effect on the boundedness since it does not make heavy tails any lighter. The regularity condition in Assumption 2 is arguably a rather general smoothness constraint. We would further conjecture that $C_\epsilon$ has order $\sigma^2 = \mu$ under suitable regularity conditions to be determined, and thus the error would be essentially $\mu^3$.

It is well-known that an infinitesimal step size is necessary in the Langevin algorithm for proper convergence, and a non-infinitesimal step size creates a bias, often of $O(\mu)$, see e.g., Vempala and Wibisono (2019). From that viewpoint, approximating Oracle Langevin with $O(\mu^3)$ could be considered very accurate and is unlikely to create much more bias. (Here, an infinitesimal step size implies that the noise level is infinitesimal, since the two are related by the condition (8).)

Admittedly, the theorem above has serious limitations, since it is only about a single step and starting from the correct distribution. Hopefully, it can serve as a starting point for a more rigorous analysis. Indeed, Beyler and Bach (2025) provide a more rigorous analysis, inspired by an earlier preprint version of our paper.

We also give a detailed analysis for the Gaussian case in Appendix E. The main additional result is that the algorithm converges, up to higher-order terms, to the same distribution as Oracle Langevin. In other words, the bias due to noisy-data score is cancelled, up to higher-order terms. Furthermore, in the Gaussian case, we can actually prove such convergence, and this convergence is global.

### 3.3 Special case: Sampling by half-denoising

A particularly interesting special case of the iteration in (6) is obtained when we set the step size

$$\mu = \frac{\sigma^2}{2} \tag{14}$$

This is the smallest $\mu$ allowed according to (8), given a noise level $\sigma^2$. In many cases, it may be useful to use this lower bound since a larger step size in Langevin methods leads to more bias.

In this case the noise $\boldsymbol{\nu}_t$ is cancelled, and we are left with a simple iteration:

$$\mathbf{x}_{t+1} = \tilde{\mathbf{x}}_t + \frac{\sigma^2}{2} \boldsymbol{\Psi}_{\tilde{\mathbf{x}}}(\tilde{\mathbf{x}}_t) \tag{15}$$

with

$$\tilde{\mathbf{x}}_t = \mathbf{x}_t + \sigma \mathbf{n}_t, \quad \mathbf{n}_t \sim \mathcal{N}(\mathbf{0}, \mathbf{I}) \tag{16}$$

where, as above, $\boldsymbol{\Psi}_{\tilde{\mathbf{x}}}$ is the score function of the noisy data and $\sigma^2$ is the variance of the noise in $\boldsymbol{\Psi}_{\tilde{\mathbf{x}}}$, typically coming from DSM.

Now, the Tweedie-Miyasawa theorem in Eq. (3) tells us that the optimal nonlinearity to reduce noise is quite similar to (15), but crucially, such optimal denoising has $\sigma^2$ as the coefficient of the score function (for isotropic noise) instead of $\sigma^2/2$ that we have here. Thus, the iteration in Eq. (15) can be called *half-denoising*.

## 4 Experiments

**Data models** We use two different scenarios. The first is *Gaussian mixture model in two dimensions*. Two dimensions was chosen so that we can easily analyze the results by kernel density estimation, as well as plot them. A GMM is a versatile family which also allows for exact sampling as a baseline. The number of kernels (components) is varied from 1 to 4; thus the simulations also include sampling from a 2D Gaussian distribution. The kernels are isotropic, and slightly overlapping. The variances of the variables in the resulting data were in the range [0.5,1.5]. We consider two different noise levels in the score function, a higher one ($\sigma^2 = 0.3$) and a lower one ($\sigma^2 = 0.1$). These are considered typical in DSM estimation.

The second scenario is *Gaussian model in higher dimensions*. The Gaussian data was white, and the dimension took the values 5,10,100. Only the higher noise level ($\sigma^2 = 0.3$) was considered.

**MCMC algorithms** The starting point is that we only know the noisy-data score function (i.e. the score function for noisy data), and for one single noise level. Using that, we sample data by the following two methods:

- "Proposed" refers to the sampling using half-denoising and the noisy-data score function as in (15-16). Assuming the noisy-data score function is estimated by DSM, this is the method we would propose to generate new data.

- "Basic Langevin" uses the ordinary Langevin method in (5), together with the noisy-data score function, since that is assumed to be the only one available. This is the (biased) baseline method that could be used for sampling based on DSM and the ordinary Langevin algorithm.

As further baselines and bases for comparison, we use:

- "Oracle Langevin", as defined above, uses the ordinary Langevin method together with the true score function $\boldsymbol{\Psi}_{\mathbf{x}}$ (i.e. of the noise-free data $\mathbf{x}$). This is against the main setting of our paper, and arguably unrealistic in deep learning where DSM is typically used. However, it is useful for analyzing the sources of errors in our method, and to validate the theory above.

- "Ground truth" means exact sampling from the true distribution. This is computed to analyze the errors stemming from the methods used in comparing samples, described below.

*Step sizes* are chosen as follows. According to the theory above, the step size for the proposed method is implicitly given based on the noise level as in (14). For the baseline Langevin methods, we use that same step size if nothing is mentioned, but conduct additional simulations with another step size which is 1/4 of the above. This is to control for the fact that the basic Langevin method has the possible advantage

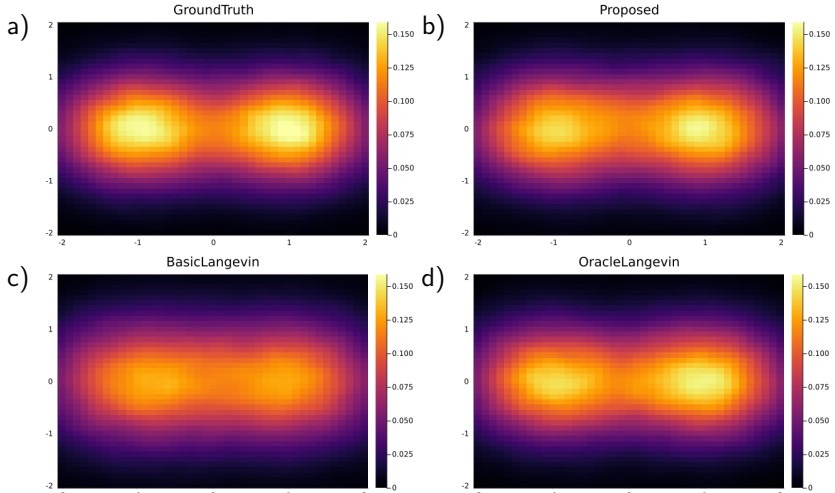

Figure 1: Kernel density estimates for some of the methods, for two kernels in GMM, and lower noise $\sigma^2 = 0.1$ in the score function. The proposed method achieves similar performance to the Oracle Langevin which uses the true score function, thus effectively cancelling the bias due to the noisy-data score function. The bias due to the non-infinitesimal step size is hardly visible.

that the step size can be freely chosen, and in particular it can be made smaller to reduce the bias due to the non-infinitesimal step size. Note that Oracle Langevin is not identical for the two noise levels precisely because its step size is here defined to be a function of the noise level. Each algorithm was run for 1,000,000 steps.

**Analysis of the sampling results**  In the main results (bias removal), the last 30% of the data was analyzed as samples of the distribution; it is assumed here that this 30% represents the final distribution and all the algorithms have converged. (The mixing analysis described next corroborates this assumption.)

In the GMM case, we compute a kernel density estimator for each sampled data set with kernel width 0.1, evaluated at a 2D grid with width of 0.1. We then compute the Euclidean ($L^2$) distance between the estimated densities. In particular, we report the distances of the MCMC method from the ground truth sample, normalized by the norm of the density of the ground truth sample, so that the errors can be intepreted as percentages (e.g. error of -1.0 in $\log_{10}$ scale means 10% error). We further evaluate the accuracy of the distance evaluation just described by computing the distance between two independent ground truth sample data sets (simply denoted by "ground truth" in the distance plots). In the high-dimensional Gaussian case, in contrast, we compute the covariance matrices and use the Euclidean distances between them as the distance measure.

Additional analyses were conducted regarding the mixing speed. We took the GMM with two kernels as above, but ran it in 10,000 trials with 100 steps in each. The algorithm was initialized as a Gaussian distribution with small variance approximately half-way between the two kernels. Here, the convergence was analyzed by looking at the Euclidean distance between the covariance of the sample and the true covariance, for each time point.

**Results 1: GMM bias removal**  Basic visualization of the results is given in Figs 1 and 2, which we will not comment any further.

A comprehensive quantitative comparison is given in Fig. 3. Most importantly, the error in the proposed method is very similar to the Oracle Langevin in the case where the step sizes are equal. This shows that almost all the bias in the method is due to the non-infinitesimal step size, as Oracle Langevin suffers from that same bias and has similar performance. Thus, removing the bias due to the noisy-data score is successful

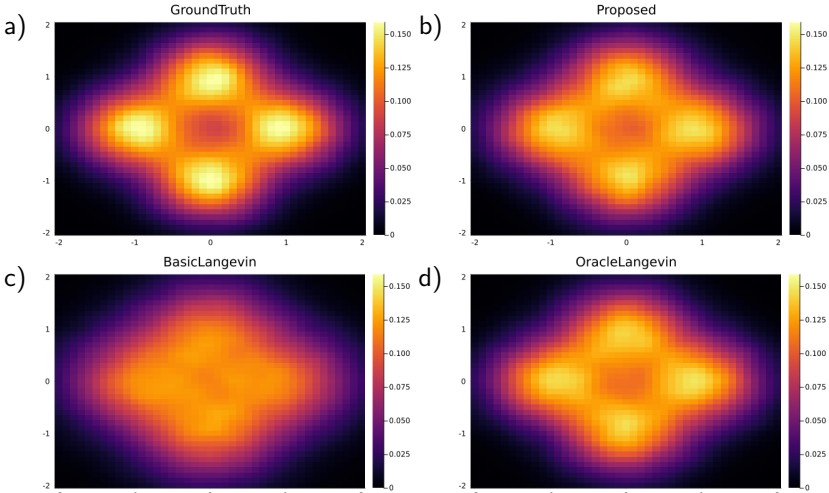

Figure 2: Like Fig. 1 above, but for four kernels in GMM, and a higher noise level $\sigma^2 = 0.3$. Now, some bias due to non-infinitesimal step size is clearly visible even in Oracle Langevin. Still, the proposed method has very similar performance to Oracle Langevin, which is the main claim we wish to demonstrate.

since any possibly remaining noise-induced bias seems to be insignificant compared to the bias induced by the step size. This corroborates the theory of this paper, in particular Theorem 2.

The basic Langevin algorithm has, in principle, the advantage that its step size can be freely chosen; however, we see that this does not give a major advantage (see curves with "mu/4") over the proposed method. In the noisy-data score case, reducing the step size improves the results of basic Langevin only very slightly; as already pointed out above, the bias due to the noisy-data score seems to be much stronger than the bias due to non-infinitesimal step size. In contrast, in the case of the Oracle Langevin, we do see that a small step size can sometimes improve results considerably, as seen especially in a), because it reduces the only bias that the method suffers from, i.e. bias due to non-infinitesimal step size. However, a very small step size may sometimes lead to worse results, as seen in b), presumably due to slow convergence or bad mixing. In any case, even Oracle Langevin fails to approach the level of the errors of "ground truth" (i.e. exact sampling, which has non-zero error due to the kernel density estimation).

**Results 2: High-dimensional Gaussian bias removal** The results, shown in Fig. 4, are very similar to the GMM case above. Again, we see that the proposed method has performance which is almost identical to the Oracle Langevin with the same step size, in line with our theory. In terms of sampling performance, the proposed method clearly beats the basic Langevin. Only the Oracle Langevin with a smaller step size is better, and only in high dimensions (arguably, it could be better in low dimensions as well if the step size were carefully tuned and/or more steps were taken).

**Results 3: Mixing speed analysis** Fig. 5 shows that the noise-corrected version mixes with equal speed to the basic Langevin. In fact, the curves are largely overlapping (e.g. solid red vs. solid blue) for the noise-corrected version and the basic Langevin, in the beginning of the algorithm. However, soon the plain Langevin plateaus due to the bias, while the proposed algorithm reduces the error further. Thus, no mixing speed seems to be lost due to the noise-correction.

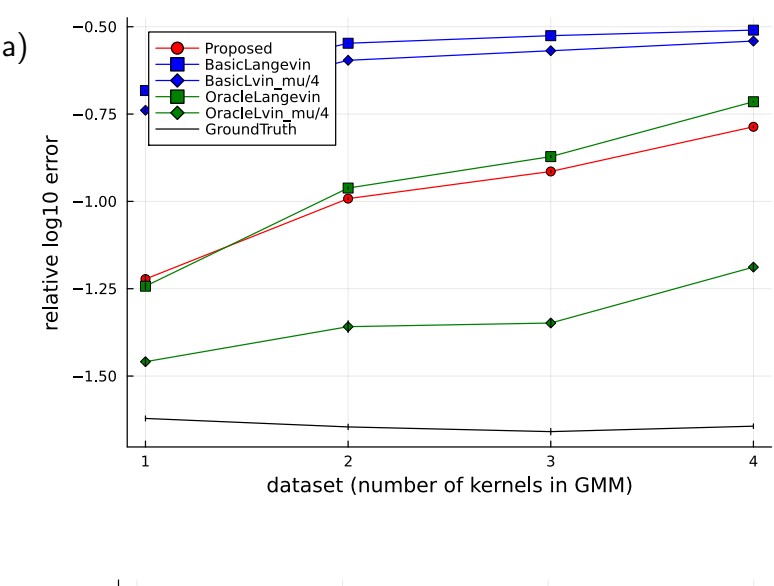

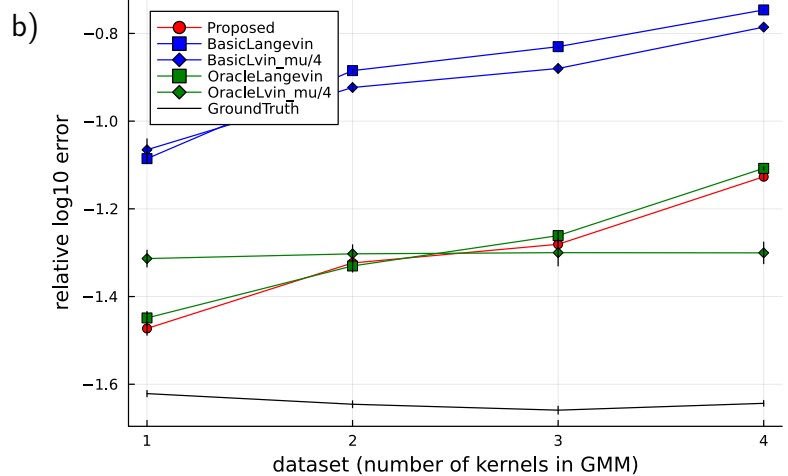

Figure 3: Gaussian 2D mixture model data: Euclidean log-distances between the kernel density estimates for the different sampling methods and the kernel density estimate of the ground truth. a) shows the results for a higher noise level ($\sigma^2 = 0.3$) and b) for a lower noise level ($\sigma^2 = 0.1$) in the score function. Error bars (often too small to be visible) give standard errors of the mean. The curve with label "Ground truth" refers to the distance between two different data sets given by exact sampling; it is far from zero due to errors in the kernel density estimation used in the distance calculation. The different plots for Langevin use different step sizes: "Basic/OracleLangevin" use the same step size as the proposed half-denoising, while the variants with "mu/4" use a step size which is one quarter of that ("Lvin" is a short for "Langevin"). Note that the step sizes for the Oracle Langevin curves are different in a) and b) although the methods are otherwise identical in the two plots; one the other hand, the ground truth is independent of the noise level and thus identical in a) and b).

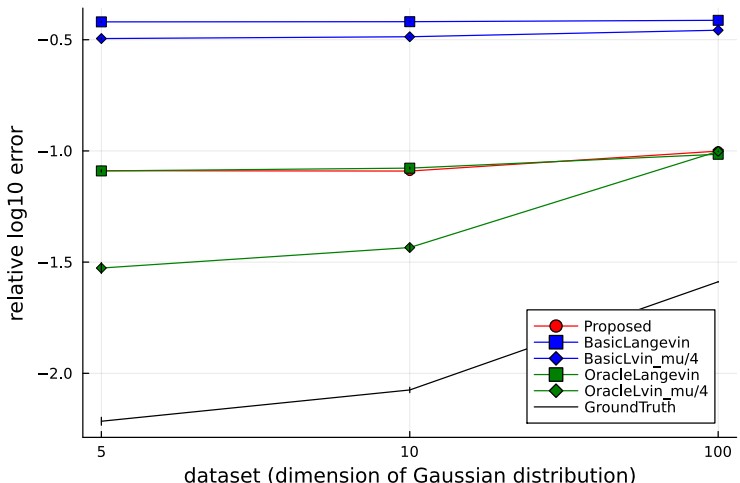

Figure 4: High-dimensional Gaussian data. Euclidean log-distances between the covariance estimates for the different sampling methods and the covariance estimate of the ground truth. Methods and legend as in Fig. 3, here shown for higher noise level ($\sigma^2 = 0.3$) only. ("Proposed" curve may be difficult to see since it is largely under the OracleLangevin curve.)

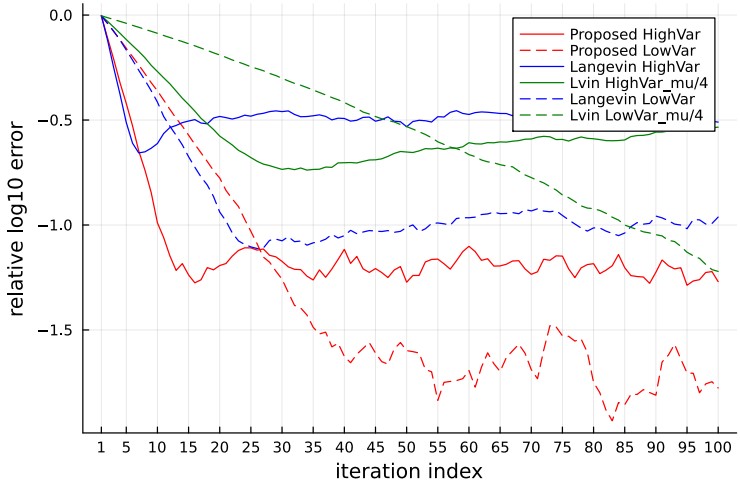

Figure 5: Analysis of mixing speed. The Euclidean log-distances (errors) as a function of iteration count, averaged over 10,000 runs, for a GMM with two kernels. Algorithm legends as in Fig. 3, but here we have the two different variance levels ("HighVar" and "LowVar"') in a single plot.

## 5 Related methods and future directions

It is often assumed that an annealed version of the Langevin algorithm is preferable in practice to avoid the algorithm get stuck in local modes (Lee et al., 2018). A method for annealing the basic Langevin method combined with score matching using different noise levels was proposed by Song and Ermon (2019) and its converge was analyzed by Guo et al. (2024). Our method could easily be combined with such annealing schedules. If the noisy-data score function were estimated for different noise levels, an annealed method would be readily obtained by starting the iterations using high noise levels and gradually moving to smaller ones. However, a special annealing method could be obtained by using a single estimate of the noisy-data score, while increasing the parameter $\sigma^2$ in the iteration, now divorced from the actual noise level in $\mathbf{\Psi_{\bar{x}}}$, from zero to $2\mu$ in the iteration in Eq. (6). We leave the utility of such annealing schedules for future research.

Saremi and Hyvärinen (2019) proposed a heuristic algorithm, called "walk-jump" sampling, inspired by the Tweedie-Miyasawa formula. Basically, they sample noisy data using ordinary Langevin, and then to obtain a sample of noise-free data, they do the full Tweedie-Miyasawa denoising. While there is no proof that this will produce unbiased samples, in further work, Frey et al. (2024); Saremi et al. (2023); Pinheiro et al. (2023) have obtained very good results in real-life application with such an iteration, mostly using an underdamped variant. A comparison of walk-jump and our method has been provided by Beyler and Bach (2025). Related methods have been proposed by Bengio et al. (2013) and Jain and Poole (2022). Likewise, diffusion models use denoising as an integral part. Connections of those methods with our method are an interesting question for future work. However, our method distinguishes itself by denoising only "half-way". Some further connection might be found between our method and underdamped Langevin methods (Dockhorn et al., 2021). In particular, an underdamped version of our method would be an interesting direction for future research.

We should also mention a strand of work that consider that the score function has noise added in it, as in finite-sample estimation of the score function (Ma et al., 2015; Zhang et al., 2022). This is of course very different from the bias due to noise added to the data, as considerered here. Another line of work, which could be relevant to develop extensions of our method, is DSM using non-Gaussian noise (Deasy et al., 2021).

## 6 Conclusion

In generative deep learning, the score function is usually estimated by DSM, which leads to a well-known bias. If the bias is not removed, any subsequent Langevin algorithm (or any ordinary MCMC) will result in biased sampling. In particular, the samples will come from the noisy distribution, that is, the original data with Gaussian noise added, which is obviously unsatisfactory in most practical applications. Therefore, we propose here how to remove that bias, i.e., how to sample from the original data distribution when only the biased ("noisy") score function learned by DSM is available. In contrast to, for example, diffusion models, only the score function for one noise level is needed. We analyze the convergence in the usual limit cases (infinite data in learning the score function, infinitesimal step size in the sampling), by showing that the algorithm is equivalent to an Oracle Langevin algorithm, i.e. Langevin used with the true score. Simple experiments on simulated data confirm the theory.

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

## A  Proof of Theorem 1 (Tweedie-Miyasawa)

We denote the pdf of $\mathbf{n}$ for notational simplicity as

$$\phi(\mathbf{n}) = \frac{1}{(2\pi)^{d/2}|\boldsymbol{\Sigma}_{\mathbf{n}}|^{1/2}} \exp(-\frac{1}{2}\mathbf{n}^T\boldsymbol{\Sigma}_{\mathbf{n}}^{-1}\mathbf{n}) \tag{17}$$

To prove the theorem, we start by the simple identity

$$p_{\tilde{\mathbf{x}}}(\tilde{\mathbf{x}}) = \int p(\tilde{\mathbf{x}}|\mathbf{x})p_{\mathbf{x}}(\mathbf{x})d\mathbf{x} = \int \phi(\tilde{\mathbf{x}} - \mathbf{x})p_{\mathbf{x}}(\mathbf{x})d\mathbf{x} \tag{18}$$

Now, we take derivatives of both sides with respect to $\tilde{\mathbf{x}}$ to obtain

$$\nabla_{\mathbf{x}}p_{\tilde{\mathbf{x}}}(\tilde{\mathbf{x}}) = \int \boldsymbol{\Sigma}_{\mathbf{n}}^{-1}(\mathbf{x} - \tilde{\mathbf{x}})\phi(\tilde{\mathbf{x}} - \mathbf{x})p_{\mathbf{x}}(\mathbf{x})d\mathbf{x} \tag{19}$$

We divide both sides by $p_{\tilde{\mathbf{x}}}(\tilde{\mathbf{x}})$ and rearrange to obtain

$$\boldsymbol{\Psi}_{\tilde{\mathbf{x}}}(\tilde{\mathbf{x}}) = \int \boldsymbol{\Sigma}_{\mathbf{n}}^{-1}\mathbf{x}\phi(\tilde{\mathbf{x}} - \mathbf{x})p_{\mathbf{x}}(\mathbf{x})/p_{\tilde{\mathbf{x}}}(\tilde{\mathbf{x}})d\mathbf{x} - \int \boldsymbol{\Sigma}_{\mathbf{n}}^{-1}\tilde{\mathbf{x}}\phi(\tilde{\mathbf{x}} - \mathbf{x})p_{\mathbf{x}}(\mathbf{x})/p_{\tilde{\mathbf{x}}}(\tilde{\mathbf{x}})d\mathbf{x}$$
$$= \boldsymbol{\Sigma}_{\mathbf{n}}^{-1}\int \mathbf{x}p(\mathbf{x}|\tilde{\mathbf{x}})d\mathbf{x} - \boldsymbol{\Sigma}_{\mathbf{n}}^{-1}\tilde{\mathbf{x}}\int p(\mathbf{x}|\tilde{\mathbf{x}})d\mathbf{x} \tag{20}$$

where the last integral is equal to unity. Multiplying both sides by $\boldsymbol{\Sigma}_{\mathbf{n}}$, and moving the last term to the left-hand-side, we get (3). □

## B  Proof of Corollary 1

Consider the general optimization problem:

$$\min_{\mathbf{f}} \mathbb{E}_{\tilde{\mathbf{x}},\mathbf{x}}\{\|\mathbf{x} - \mathbf{f}(\tilde{\mathbf{x}})\|^2\|\} \tag{21}$$

and denote the optimal $\mathbf{f}$ by $\mathbf{f}^*$. By well-known properties of the conditional expectation,

$$\mathbf{f}^*(\tilde{\mathbf{x}}) = \mathbb{E}\{\mathbf{x}|\tilde{\mathbf{x}}\} \tag{22}$$

Now, considering the optimization in the Corollary:

$$\min_{\boldsymbol{\Psi}} \mathbb{E}_{\tilde{\mathbf{x}},\mathbf{x}}\{\|\mathbf{x} - (\tilde{\mathbf{x}} + \boldsymbol{\Sigma}_{\mathbf{n}}\boldsymbol{\Psi}(\tilde{\mathbf{x}}))\|^2\} \tag{23}$$

and denoting the optimal $\boldsymbol{\Psi}$ by $\boldsymbol{\Psi}^*$, we find that

$$\tilde{\mathbf{x}} + \boldsymbol{\Sigma_n}\boldsymbol{\Psi}^*(\tilde{\mathbf{x}}) = \mathbb{E}\{\mathbf{x}|\tilde{\mathbf{x}}\} \tag{24}$$

On the other hand, by the Tweedie-Miyasawa Theorem, we have

$$\mathbb{E}\{\mathbf{x}|\tilde{\mathbf{x}}\} = \tilde{\mathbf{x}} + \boldsymbol{\Sigma_n}\boldsymbol{\Psi}_{\tilde{\mathbf{x}}}(\tilde{\mathbf{x}}) \tag{25}$$

and combining these implies $\boldsymbol{\Psi}^* = \boldsymbol{\Psi}_{\tilde{\mathbf{x}}}$. Thus, solving the self-supervised learning problem in the Corollary estimates the score $\boldsymbol{\Psi}_{\tilde{\mathbf{x}}}$ of the noisy data $\tilde{\mathbf{x}}$.

## C  Main proof of Theorem 2

Define a random variable as

$$\mathbf{y}_t = \tilde{\mathbf{x}}_t + \mu\boldsymbol{\Psi}_{\tilde{\mathbf{x}}}(\tilde{\mathbf{x}}_t) \tag{26}$$

Assume we are at a point where $\mathbf{x}_t \sim \mathbf{x}$ and thus also $\tilde{\mathbf{x}}_t \sim \tilde{\mathbf{x}}$. We can drop the index $t$ for simplicity. This confounds the $\mathbf{x}_t$ in the algorithm and $\mathbf{x}$ as in the original data being modelled (and likewise for the noisy case), but the distributions are the same by this assumption so this is not a problem.

Consider the characteristic function (Fourier transform of pdf) of $\mathbf{y}$, and make a first-order approximation as:

$$\hat{p}_{\mathbf{y}}(\boldsymbol{\xi}) = \mathbb{E}_{\mathbf{y}}\exp(i\boldsymbol{\xi}^T\mathbf{y}) = \mathbb{E}_{\tilde{\mathbf{x}}}\exp(i\boldsymbol{\xi}^T\tilde{\mathbf{x}})\exp(\mu i\boldsymbol{\xi}^T\boldsymbol{\Psi}_{\tilde{\mathbf{x}}}(\tilde{\mathbf{x}}))$$

$$= \mathbb{E}_{\tilde{\mathbf{x}}}\exp(i\boldsymbol{\xi}^T\tilde{\mathbf{x}})[1 + \mu i\boldsymbol{\xi}^T\boldsymbol{\Psi}_{\tilde{\mathbf{x}}}(\tilde{\mathbf{x}}) - \frac{1}{2}\mu^2(\boldsymbol{\xi}^T\boldsymbol{\Psi}_{\tilde{\mathbf{x}}}(\tilde{\mathbf{x}}))^2] + O(\mu^3 C_{\Psi}^2\|\boldsymbol{\xi}\|^3) \tag{27}$$

where the last equality holds, especially regarding its higher-order terms, under the assumption of bounded score (Assumption 1), as proven in Lemma 1 below. Now, we analyze the terms in this expansion.

**First terms in RHS of (27)**  Elementary manipulations further give for the first two terms in brackets, multiplied by what is multiplying the brackets:

$$\mathbb{E}_{\tilde{\mathbf{x}}}\exp(i\boldsymbol{\xi}^T\tilde{\mathbf{x}})[1 + \mu i\boldsymbol{\xi}^T\boldsymbol{\Psi}_{\tilde{\mathbf{x}}}(\tilde{\mathbf{x}})] = \hat{p}_{\tilde{\mathbf{x}}}(\boldsymbol{\xi}) + \mu i\int\exp(i\boldsymbol{\xi}^T\tilde{\mathbf{x}})\sum_{j=1}^k\xi_j\frac{\partial p_{\tilde{\mathbf{x}}}(\tilde{\mathbf{x}})}{\partial\tilde{x}_j}d\tilde{\mathbf{x}} \tag{28}$$

Now, we use the well-known trick of integration by parts, and we obtain

$$\sum_{j=1}^k\int\exp(i\boldsymbol{\xi}^T\tilde{\mathbf{x}})\xi_j\frac{\partial p_{\tilde{\mathbf{x}}}(\tilde{\mathbf{x}})}{\partial\tilde{x}_j}d\tilde{\mathbf{x}} = -\sum_{j=1}^k i\xi_j^2\int\exp(i\boldsymbol{\xi}^T\tilde{\mathbf{x}})p_{\tilde{\mathbf{x}}}(\tilde{\mathbf{x}})d\tilde{\mathbf{x}} = -i\|\boldsymbol{\xi}\|^2\hat{p}_{\tilde{\mathbf{x}}}(\boldsymbol{\xi}) \tag{29}$$

**Last terms in RHS of (27)**  Regarding the last term in (27), elementary manipulations give

$$\mathbb{E}_{\tilde{\mathbf{x}}}\exp(i\boldsymbol{\xi}^T\tilde{\mathbf{x}})(\boldsymbol{\xi}^T\boldsymbol{\Psi}_{\tilde{\mathbf{x}}}(\tilde{\mathbf{x}}))^2 = \boldsymbol{\xi}^T[\int\exp(i\boldsymbol{\xi}^T\mathbf{z})\boldsymbol{\Psi}_{\tilde{\mathbf{x}}}(\mathbf{z})\boldsymbol{\Psi}_{\tilde{\mathbf{x}}}(\mathbf{z})^T p_{\tilde{\mathbf{x}}}(\mathbf{z})d\mathbf{z}]\boldsymbol{\xi} \tag{30}$$

Let us denote

$$\epsilon(\mathbf{z}) = p_{\tilde{\mathbf{x}}}(\mathbf{z}) - p_{\mathbf{x}}(\mathbf{z}) \tag{31}$$

and

$$\boldsymbol{\Psi}_\epsilon(\mathbf{z}) = \boldsymbol{\Psi}_{\tilde{\mathbf{x}}}(\mathbf{z}) - \boldsymbol{\Psi}_{\mathbf{x}}(\mathbf{z}) \tag{32}$$

which are related to the bounding terms in 13. Now, consider simply the trace to see the order of the terms, and expand the matrix above as:

$$\mathrm{tr}(\int\exp(i\boldsymbol{\xi}^T\mathbf{z})\boldsymbol{\Psi}_{\tilde{\mathbf{x}}}(\mathbf{z})\boldsymbol{\Psi}_{\tilde{\mathbf{x}}}(\mathbf{z})^T p_{\tilde{\mathbf{x}}}(\mathbf{z})d\mathbf{z}) = \int\exp(i\boldsymbol{\xi}^T\mathbf{z})\boldsymbol{\Psi}_{\mathbf{x}}(\mathbf{z})^T\boldsymbol{\Psi}_{\mathbf{x}}(\mathbf{z})p_{\mathbf{x}}(\mathbf{z})d\mathbf{z}$$

$$+ \int\exp(i\boldsymbol{\xi}^T\mathbf{z})\boldsymbol{\Psi}_{\mathbf{x}}(\mathbf{z})^T\boldsymbol{\Psi}_{\mathbf{x}}(\mathbf{z})\epsilon(\mathbf{z})d\mathbf{z}$$

$$+ 2\int\exp(i\boldsymbol{\xi}^T\mathbf{z})\boldsymbol{\Psi}_{\tilde{\mathbf{x}}}(\mathbf{z})^T\boldsymbol{\Psi}_\epsilon(\mathbf{z})p_{\tilde{\mathbf{x}}}(\mathbf{z})d\mathbf{z} + \int\exp(i\boldsymbol{\xi}^T\mathbf{z})\boldsymbol{\Psi}_\epsilon(\mathbf{z})^T\boldsymbol{\Psi}_\epsilon(\mathbf{z})p_{\tilde{\mathbf{x}}}(\mathbf{z})d\mathbf{z} \tag{33}$$

Consider now that the $\boldsymbol{\Psi_x}$ is bounded by assumption, and by Lemma 2, $\boldsymbol{\Psi_{\tilde{x}}}$ is bounded by the same constant $C_\Psi$. Looking at the RHS, the second and third terms are of the order of the two terms defining $C_\epsilon$ in (13), while the fourth term is of lower order (with $\epsilon^2$). Thus, the residual terms coming from omitting the other terms than the first one in (33), considering that they are multiplied by $\mu^2$ in (27), become $O(\mu^2 C_\epsilon \|\boldsymbol{\xi}\|^2)$.

Denote the matrix integral in the first term by $\mathbf{M}(\boldsymbol{\xi}) = \int \exp(i\boldsymbol{\xi}^T\mathbf{z})\boldsymbol{\Psi_x}(\mathbf{z})\boldsymbol{\Psi_x}(\mathbf{z})^T p_\mathbf{x}(\mathbf{z})d\mathbf{z}$; this matrix is bounded by assumption of bounded score.

**Putting the results together**   Thus we have proven the following form for (27)

$$\hat{p}_\mathbf{y}(\boldsymbol{\xi}) = \hat{p}_{\tilde{\mathbf{x}}}(\boldsymbol{\xi})(1 + \mu\|\boldsymbol{\xi}\|^2) - \frac{1}{2}\mu^2\boldsymbol{\xi}^T\mathbf{M}(\boldsymbol{\xi})\boldsymbol{\xi} + O(\max(\mu^3 C_\Psi^3\|\boldsymbol{\xi}\|^3, \mu^2 C_\epsilon\|\boldsymbol{\xi}\|^2))$$

(34)

Using the well-known formula for the characteristic function of Gaussian isotropic noise of variance $\sigma^2$, we have:

$$\hat{p}_{\tilde{\mathbf{x}}}(\boldsymbol{\xi}) = \hat{p}_\mathbf{x}(\boldsymbol{\xi})\exp(-\frac{1}{2}\sigma^2\|\boldsymbol{\xi}\|^2)$$

(35)

Thus, we can calculate the characteristic function of $\mathbf{x}_{t+1}$, which we denote by $\hat{p}_+$, by multiplying $\hat{p}_\mathbf{y}$ by the characteristic function of the Gaussian isotropic noise with variance $2\mu - \sigma^2$, and obtain

$$\hat{p}_+(\boldsymbol{\xi}) = \hat{p}_\mathbf{y}(\boldsymbol{\xi})\exp(-\frac{1}{2}(2\mu - \sigma^2)\|\boldsymbol{\xi}\|^2)$$

$$= [\hat{p}_\mathbf{x}(\boldsymbol{\xi})\exp(-\frac{1}{2}\sigma^2\|\boldsymbol{\xi}\|^2)(1+\mu\|\boldsymbol{\xi}\|^2) - \frac{1}{2}\mu^2\boldsymbol{\xi}^T\mathbf{M}(\boldsymbol{\xi})\boldsymbol{\xi}]\exp(-\frac{1}{2}(2\mu-\sigma^2)\|\boldsymbol{\xi}\|^2) + O(\max(\mu^3 C_\Psi^3\|\boldsymbol{\xi}\|^3, \mu^2 C_\epsilon\|\boldsymbol{\xi}\|^2))$$

$$= \hat{p}_\mathbf{x}(\boldsymbol{\xi})\exp(-\mu\|\boldsymbol{\xi}\|^2)(1 + \mu\|\boldsymbol{\xi}\|^2) - \frac{1}{2}\mu^2\boldsymbol{\xi}^T\mathbf{M}(\boldsymbol{\xi})\boldsymbol{\xi}\exp(-\frac{1}{2}(2\mu - \sigma^2)\|\boldsymbol{\xi}\|^2) + O(\max(\mu^3 C_\Psi^3\|\boldsymbol{\xi}\|^3, \mu^2 C_\epsilon\|\boldsymbol{\xi}\|^2))$$

$$= \hat{p}_\mathbf{x}(\boldsymbol{\xi})\exp(-\mu\|\boldsymbol{\xi}\|^2)(1 + \mu\|\boldsymbol{\xi}\|^2) - \frac{1}{2}\mu^2\boldsymbol{\xi}^T\mathbf{M}(\boldsymbol{\xi})\boldsymbol{\xi} + O(\max(\mu^3 C_\Psi^3\|\boldsymbol{\xi}\|^3, \mu^3\|\boldsymbol{\xi}\|^2, \mu^2 C_\epsilon\|\boldsymbol{\xi}\|^2))$$   (36)

where in the last equality, we have got rid of the remaining exponential term since it is equal to $1 + O(\mu)$, and the $O(\mu)$ resilts as terms of $O(\mu^3\|\boldsymbol{\xi}\|^2)$ in $O$ term.

Now, the terms with $\sigma^2$ have disappeared, and no noisy-data score is left in the equation (except in the higher-order terms). So, the iteration is always equal to the same iteration with $\sigma^2 = 0$ and the original score, up to the higher-order terms. But such an iteration is exactly the Oracle Langevin iteration. Thus, one iteration in the algorithm is equal to one iteration of Oracle Langevin, up to the higher-order terms given above, if the algorithm is at the true distribution of $p_\mathbf{x}$.□

# D   Lemmas used in Theorem 2

Next, we prove the two Lemmas used in the in the proof of Theorem 2.

**Lemma 1** *Assume that the norm of the score $\|\boldsymbol{\Psi_x}\|$ is bounded by $C_\Psi$. Then, the first-order approximation in (27) has error $O(\mu^3 C_\Psi\|\boldsymbol{\xi}\|^3)$ in the limit of infinitesimal $\mu$.*

Proof of Lemma: The sum of the terms in the Taylor expansion of order higher than two, i.e. omitted in the approximation, is

$$\sum_{m=3}^\infty \frac{(\mu i)^m}{m!}(\boldsymbol{\xi}^T\boldsymbol{\Psi}_{\tilde{\mathbf{x}}}(\tilde{\mathbf{x}}))^m$$

(37)

which, when combined with the expectation gives the remainder term, denote it by $\alpha$, as

$$\alpha = \mathbb{E}_{\tilde{\mathbf{x}}}\exp(i\boldsymbol{\xi}^T\tilde{\mathbf{x}})\left[\sum_{m=3}^\infty \frac{(\mu i)^m}{m!}(\boldsymbol{\xi}^T\boldsymbol{\Psi}_{\tilde{\mathbf{x}}}(\tilde{\mathbf{x}}))^m\right] = \sum_{m=3}^\infty \frac{(\mu i)^m}{m!}\mathbb{E}_{\tilde{\mathbf{x}}}\exp(i\boldsymbol{\xi}^T\tilde{\mathbf{x}})(\boldsymbol{\xi}^T\boldsymbol{\Psi}_{\tilde{\mathbf{x}}}(\tilde{\mathbf{x}}))^m$$

(38)

This can be bounded using elementary inequalities as follows:

$$|\alpha| \leq \sum_{m=3}^{\infty} \frac{\mu^m}{m!} |\mathbb{E}_{\tilde{\mathbf{x}}} \exp(i\boldsymbol{\xi}^T \tilde{\mathbf{x}})(\boldsymbol{\xi}^T \boldsymbol{\Psi}_{\tilde{\mathbf{x}}}(\tilde{\mathbf{x}}))^m| \leq \sum_{m=3}^{\infty} \frac{\mu^m}{m!} \mathbb{E}_{\tilde{\mathbf{x}}} |\exp(i\boldsymbol{\xi}^T \tilde{\mathbf{x}})||\boldsymbol{\xi}^T \boldsymbol{\Psi}_{\tilde{\mathbf{x}}}(\tilde{\mathbf{x}})|^m$$

$$= \sum_{m=3}^{\infty} \frac{\mu^m}{m!} \mathbb{E}_{\tilde{\mathbf{x}}} |\boldsymbol{\xi}^T \boldsymbol{\Psi}_{\tilde{\mathbf{x}}}(\tilde{\mathbf{x}})|^m \leq \sum_{m=3}^{\infty} \frac{\mu^m}{m!} \mathbb{E}_{\tilde{\mathbf{x}}} (\|\boldsymbol{\xi}\| \|\boldsymbol{\Psi}_{\tilde{\mathbf{x}}}(\tilde{\mathbf{x}})\|)^m = \sum_{m=3}^{\infty} \frac{\mu^m}{m!} \|\boldsymbol{\xi}\|^m \mathbb{E}_{\tilde{\mathbf{x}}} \|\boldsymbol{\Psi}_{\tilde{\mathbf{x}}}(\tilde{\mathbf{x}})\|^m$$

$$\leq \sum_{m=3}^{\infty} \frac{1}{m!} (\mu C_\Psi \|\boldsymbol{\xi}\|)^m = f(\mu C_\Psi \|\boldsymbol{\xi}\|) \quad (39)$$

where in the last inequality we have used Lemma 2, which says that $\boldsymbol{\Psi}_{\tilde{\mathbf{x}}}$ is bounded by the same $C_\Psi$ as $\boldsymbol{\Psi}_{\mathbf{x}}$. We also see that the function

$$f(z) = \exp(z) - 1 - z - \frac{1}{2}z^2 \quad (40)$$

is increasing and it vanishes at zero where its first and second derivatives vanish as well. Thus, for small enough $\mu$, it behaves cubically, and our Taylor approximation is valid. $\square$

**Lemma 2** *If $\boldsymbol{\Psi}_{\mathbf{x}}$ is bounded, so is $\boldsymbol{\Psi}_{\tilde{\mathbf{x}}}$, and by the same constant.*

Proof: We first prove

$$\boldsymbol{\Psi}_{\tilde{\mathbf{x}}}(\mathbf{y}) = \mathbb{E}\{\boldsymbol{\Psi}_{\mathbf{x}}(\mathbf{x}) | \tilde{\mathbf{x}} = \mathbf{y}\} \quad (41)$$

which is perhaps well-known, and can be proven as follows. We start by a simple identity:

$$\log p(\tilde{\mathbf{x}}, \mathbf{x}) = \log p(\tilde{\mathbf{x}}|\mathbf{x}) + \log p_{\mathbf{x}}(\mathbf{x}) = \log \phi(\tilde{\mathbf{x}} - \mathbf{x}) + \log p_{\mathbf{x}}(\mathbf{x}) \quad (42)$$

Next, we take the derivatives of both sides with respect to $\mathbf{x}$, as opposed to $\tilde{\mathbf{x}}$ in the proof of Miyasawa-Tweedie. We obtain

$$\nabla_{\mathbf{x}} \log p(\tilde{\mathbf{x}}, \mathbf{x}) = \boldsymbol{\Sigma}_{\mathbf{n}}^{-1}(\tilde{\mathbf{x}} - \mathbf{x}) + \nabla_{\mathbf{x}} \log p_{\mathbf{x}}(\mathbf{x}) \quad (43)$$

We multiply both sides by $p(\mathbf{x}|\tilde{\mathbf{x}})$ and integrate over $\mathbf{x}$:

$$\int (\nabla_{\mathbf{x}} \log p(\tilde{\mathbf{x}}, \mathbf{x})) p(\mathbf{x}|\tilde{\mathbf{x}}) d\mathbf{x} = \boldsymbol{\Sigma}_{\mathbf{n}}^{-1}[\tilde{\mathbf{x}} - \mathbb{E}\{\mathbf{x}|\tilde{\mathbf{x}}\}] + \mathbb{E}\{\nabla_{\mathbf{x}} \log p_{\mathbf{x}}(\mathbf{x})|\tilde{\mathbf{x}}\} \quad (44)$$

Applying the Miyasawa-Tweedie theorem, we see that the term in brackets becomes

$$\tilde{\mathbf{x}} - \mathbb{E}\{\mathbf{x}|\tilde{\mathbf{x}}\} = \boldsymbol{\Sigma}_{\mathbf{n}} \boldsymbol{\Psi}_{\tilde{\mathbf{x}}}(\tilde{\mathbf{x}}) \quad (45)$$

Therefore, taking account of the fact that $\nabla_{\mathbf{x}} \log p(\tilde{\mathbf{x}}, \mathbf{x}) = \nabla_{\mathbf{x}} \log p(\mathbf{x}|\tilde{\mathbf{x}})$, (44) becomes

$$\int (\nabla_{\mathbf{x}} \log p(\mathbf{x}|\tilde{\mathbf{x}})) p(\mathbf{x}|\tilde{\mathbf{x}}) d\mathbf{x} = \boldsymbol{\Psi}_{\tilde{\mathbf{x}}}(\tilde{\mathbf{x}}) - \mathbb{E}\{\boldsymbol{\Psi}_{\mathbf{x}}(\mathbf{x})|\tilde{\mathbf{x}}\} \quad (46)$$

Regarding the left-hand side, we note it is the expectation of the gradient of the log-pdf, which is well known to be always zero. To see this, consider for simplicity the integral without conditioning to obtain

$$\int (\nabla_{\mathbf{x}} \log p(\mathbf{x})) p(\mathbf{x}) d\mathbf{x} = \int \nabla_{\mathbf{x}} p(\mathbf{x}) d\mathbf{x} = 0. \quad (47)$$

Thus, we have proven (41). Now, if $\boldsymbol{\Psi}_{\mathbf{x}}$ on the RHS is bounded, so is its (conditional) expectation, and by the same constant, which proves the Lemma. $\square$

## E   Analysis of the Gaussian case

Let us consider the Gaussian case. Assume the algorithm is initialized at a Gaussian distribution, and we run the algorithm with some noisy Gaussian score:

$$\boldsymbol{\Psi}_{\tilde{\mathbf{x}}}(\boldsymbol{\xi}) = -\boldsymbol{\Sigma}_{\tilde{\mathbf{x}}}^{-1}\boldsymbol{\xi} \tag{48}$$

Denote for simplicity $\mathbf{x}_t$ by $\mathbf{x}$ (and same with $\boldsymbol{\nu}$ and $\mathbf{n}$) and $\mathbf{x}_{t+1}$ by $\mathbf{x}'$. The iteration becomes

$$\mathbf{x}' = \tilde{\mathbf{x}} + \mu\boldsymbol{\Psi}_{\tilde{\mathbf{x}}}(\tilde{\mathbf{x}}) + \sqrt{2\mu - \sigma^2}\boldsymbol{\nu} = \mathbf{x} + \sigma\mathbf{n} - \mu\boldsymbol{\Sigma}_{\tilde{\mathbf{x}}}^{-1}(\mathbf{x} + \sigma\mathbf{n}) + \sqrt{2\mu - \sigma^2}\boldsymbol{\nu}$$
$$= [\mathbf{I} - \mu\boldsymbol{\Sigma}_{\tilde{\mathbf{x}}}^{-1}]\mathbf{x} + \sigma[\mathbf{I} - \mu\boldsymbol{\Sigma}_{\tilde{\mathbf{x}}}^{-1}]\mathbf{n} + \sqrt{2\mu - \sigma^2}\boldsymbol{\nu} \tag{49}$$

This iteration stays Gaussian and zero-mean, when started at such a distribution, which we assume in the following. So, we only need to analyze the covariance. Denote

$$\mathbf{M} = [\mathbf{I} - \mu\boldsymbol{\Sigma}_{\tilde{\mathbf{x}}}^{-1}] \tag{50}$$

Denoting the covariance of the sampled $\mathbf{x}$ by $\boldsymbol{\Sigma}$, the covariance $\boldsymbol{\Sigma}_{\mathbf{x}'} := \text{cov}(\mathbf{x}')$ can be calculated as

$$\boldsymbol{\Sigma}_{\mathbf{x}'} = \mathbf{M}\boldsymbol{\Sigma}\mathbf{M} + \sigma^2\mathbf{M}^2 + (2\mu - \sigma^2)\mathbf{I} \tag{51}$$

To solve a fixed-point of this iteration, consider the EVD

$$\mathbf{M} = \mathbf{U}\mathbf{D}_{\mathbf{M}}\mathbf{U}^T \tag{52}$$

and assume that the covariance of the sampled $\mathbf{x}$, or $\boldsymbol{\Sigma}$, can be expressed in the same eigenvectors as

$$\boldsymbol{\Sigma} = \mathbf{U}\mathbf{D}_{\mathbf{x}}\mathbf{U}^T \tag{53}$$

so the iteration in (51) becomes

$$\boldsymbol{\Sigma}_{\mathbf{x}'} = \mathbf{U}\mathbf{D}_{\mathbf{M}}^2\mathbf{D}_{\mathbf{x}}\mathbf{U}^T + \sigma^2\mathbf{U}\mathbf{D}_{\mathbf{M}}^2\mathbf{U}^T + (2\mu - \sigma^2)\mathbf{U}\mathbf{U}^T \tag{54}$$

and we see that $\boldsymbol{\Sigma}_{\mathbf{x}'}$ can be expressed with the same eigenvectors, and the iteration only consists of the diagonal entries

$$\mathbf{D}_{\mathbf{x}'} = \mathbf{D}_{\mathbf{M}}^2\mathbf{D}_{\mathbf{x}} + \sigma^2\mathbf{D}_{\mathbf{M}}^2 + (2\mu - \sigma^2)\mathbf{I} \tag{55}$$

which is stationary if

$$\mathbf{D}_{\mathbf{x}} = [\mathbf{I} - \mathbf{D}_{\mathbf{M}}^2]^{-1}[\sigma^2\mathbf{D}_{\mathbf{M}}^2 + (2\mu - \sigma^2)\mathbf{I}] \tag{56}$$

and we can go back to the original matrices and see that the covariance at the fixed point fulfills

$$\boldsymbol{\Sigma} = [\mathbf{I} - \mathbf{M}^2]^{-1}[\sigma^2\mathbf{M}^2 + (2\mu - \sigma^2)\mathbf{I}] \tag{57}$$

which can be expanded by plugging in the definition of $\mathbf{M}$ as

$$\boldsymbol{\Sigma} = [2\mu\boldsymbol{\Sigma}_{\tilde{\mathbf{x}}}^{-1} - \mu^2(\boldsymbol{\Sigma}_{\tilde{\mathbf{x}}}^{-1})^2]^{-1}[\sigma^2(\mathbf{I} - 2\mu\boldsymbol{\Sigma}_{\tilde{\mathbf{x}}}^{-1} + \mu^2(\boldsymbol{\Sigma}_{\tilde{\mathbf{x}}}^{-1})^2) + (2\mu - \sigma^2)\mathbf{I}]$$
$$= [2\boldsymbol{\Sigma}_{\tilde{\mathbf{x}}}^{-1} - \mu(\boldsymbol{\Sigma}_{\tilde{\mathbf{x}}}^{-1})^2]^{-1}[-2\sigma^2\boldsymbol{\Sigma}_{\tilde{\mathbf{x}}}^{-1} + \mu\sigma^2(\boldsymbol{\Sigma}_{\tilde{\mathbf{x}}}^{-1})^2 + 2\mathbf{I}] = -\sigma^2\mathbf{I} + [\boldsymbol{\Sigma}_{\tilde{\mathbf{x}}}^{-1} - \frac{\mu}{2}(\boldsymbol{\Sigma}_{\tilde{\mathbf{x}}}^{-1})^2]^{-1}$$
$$= -\sigma^2\mathbf{I} + \boldsymbol{\Sigma}_{\tilde{\mathbf{x}}}[\mathbf{I} - \frac{\mu}{2}(\boldsymbol{\Sigma}_{\tilde{\mathbf{x}}}^{-1})]^{-1} \tag{58}$$

At this point, note that a) the above calculations were exact, and b) the $\sigma^2$ expresses solely the noise added in the iteration of the algorithm, while $\boldsymbol{\Sigma}_{\tilde{\mathbf{x}}}^{-1}$ is the score used in the algorithm, possibly for a different (incorrect) noise level. While in the definition of our algorithm the noise level in $\boldsymbol{\Sigma}_{\tilde{\mathbf{x}}}^{-1}$ is equal to $\sigma^2$, this equation does not assume such equality, which will be useful in further analysis below.

Now assume $\mathbf{\Sigma}_{\tilde{\mathbf{x}}}^{-1}$ is actually the score of the noisy data as in the original specification of the algorithm: $\mathbf{\Sigma}_{\tilde{\mathbf{x}}} = \mathbf{\Sigma}_{\mathbf{x}} + \sigma^2 \mathbf{I}$. We make a Taylor expansion for the matrix inverse

$$\mathbf{\Sigma} = -\sigma^2 \mathbf{I} + \mathbf{\Sigma}_{\tilde{\mathbf{x}}}[\mathbf{I} + \frac{\mu}{2}\mathbf{\Sigma}_{\tilde{\mathbf{x}}}^{-1} + \frac{\mu^2}{4}(\mathbf{\Sigma}_{\tilde{\mathbf{x}}}^{-1})^2 + o(\mu^2)] = -\sigma^2 \mathbf{I} + (\mathbf{\Sigma}_{\mathbf{x}} + \sigma^2 \mathbf{I}) + \frac{\mu}{2}\mathbf{I} + \frac{\mu^2}{4}(\mathbf{\Sigma}_{\tilde{\mathbf{x}}}^{-1}) + o(\mu^2)$$
$$= \mathbf{\Sigma}_{\mathbf{x}} + \frac{\mu}{2}\mathbf{I} + O(\mu^2) \quad (59)$$

Here, we see that the error (bias) in the covariance of the converged sample is $\frac{\mu}{2}\mathbf{I}$ for our algorithm. Such a bias is expected due to finite step size. Next, we compare this error with original Langevin and Oracle Langevin cases.

**Original Langevin (misspecified)** The developments above are valid up to (58) for the original Langevin iteration, which would here be misspecified in the sense of being applied using the noisy-data score function. (As noted, the effects of misspecification of the score function and the noise-correction in the algorithm are separated in the developments above.) In particular, if $\sigma^2$, corresponding to the noise-correction in that equation, is set to zero, we get the original Langevin. Then we have, when $\mathbf{\Sigma}_{\tilde{\mathbf{x}}}$ is expanded so that in the score function, the noise is not zero:

$$\mathbf{\Sigma} = \mathbf{\Sigma}_{\tilde{\mathbf{x}}}[\mathbf{I} + \frac{\mu}{2}\mathbf{\Sigma}_{\tilde{\mathbf{x}}}^{-1} + \frac{\mu^2}{4}(\mathbf{\Sigma}_{\tilde{\mathbf{x}}}^{-1})^2 + o(\mu^2)] = (\mathbf{\Sigma}_{\mathbf{x}} + \sigma^2 \mathbf{I}) + \frac{\mu}{2}\mathbf{I} + \frac{\mu^2}{4}(\mathbf{\Sigma}_{\tilde{\mathbf{x}}}^{-1}) + o(\mu^2)$$
$$= \mathbf{\Sigma}_{\mathbf{x}} + [\sigma^2 + \frac{\mu}{2}]\mathbf{I} + O(\mu^2) \quad (60)$$

where we see that the error is greater than in our noise-corrected algorithm. Considering the step size equal to half-denoising, i.e. $\mu = \sigma^2/2$, we have

$$\sigma^2 + \frac{\mu}{2} = 2\mu + \frac{\mu}{2} = 5 \times \frac{\mu}{2} \quad (61)$$

i.e., the error in the original Langevin is exactly *five times* as large as in our algorithm. In other words, the error coming from misspesification of the score function is four times the error due to the non-infinitesimal step size.

**Oracle Langevin** The developments above are again valid up to (58) for Oracle Langevin, if $\sigma^2$ in that equation is set to zero, and further we set $\mathbf{\Sigma}_{\tilde{\mathbf{x}}} = \mathbf{\Sigma}_{\mathbf{x}}$ in the score function. Then we have

$$\mathbf{\Sigma} = \mathbf{\Sigma}_{\tilde{\mathbf{x}}}[\mathbf{I} + \frac{\mu}{2}\mathbf{\Sigma}_{\tilde{\mathbf{x}}}^{-1} + \frac{\mu^2}{4}(\mathbf{\Sigma}_{\tilde{\mathbf{x}}}^{-1})^2 + o(\mu^2)] = \mathbf{\Sigma}_{\mathbf{x}} + \frac{\mu}{2}\mathbf{I} + \frac{\mu^2}{4}(\mathbf{\Sigma}_{\tilde{\mathbf{x}}}^{-1}) + o(\mu^2) = \mathbf{\Sigma}_{\mathbf{x}} + \frac{\mu}{2}\mathbf{I} + O(\mu^2) \quad (62)$$

where we see that the error is *exactly the same* as in our noise-corrected version (up to second-order terms in $\mu$). This further corroborates our claim on bias removal.

**Convergence analysis** Finally, we proceed to a brief convergence analysis in the Gaussian case. Suppose $\mathbf{\Sigma}_0$ is a stationary point of the iteration in (51). Perturb it as $\mathbf{\Sigma}_0 + \boldsymbol{\epsilon}$. It is trivial to see that due to the linearity of (51), the iteration for the perturbation becomes

$$\boldsymbol{\epsilon}' = \mathbf{M}\boldsymbol{\epsilon}\mathbf{M} \quad (63)$$

and we only need to prove that this converges to zero. Consider the spectral norm of the matrices. We have

$$\|\boldsymbol{\epsilon}'\| \leq \|\mathbf{M}\|\|\boldsymbol{\epsilon}\|\|\mathbf{M}\| \quad (64)$$

so we only need to prove that the spectral norm of $\mathbf{M}$ is less than one. But $\mathbf{\Sigma}_{\tilde{\mathbf{x}}}^{-1}$ is positive definite, and thus, for a small enough $\mu$, $\mathbf{M} = \mathbf{I} - \mu\mathbf{\Sigma}_{\tilde{\mathbf{x}}}^{-1}$ is positive definite and has spectral norm less than one. Thus, for any perturbation, not necessarily infinitesimal, the algorithm will converge to the same point, which was found above. This proves the global convergence of the algorithm in the case of Gaussian covariances. The assumptions made were that a) the step size is small enough, and b) the algorithm is initialized as Gaussian (for example initialization at zero, i.e. Gaussian with zero variance, is also valid).

