# OpenReview forum: "A noise-corrected Langevin algorithm and sampling by half-denoising"
_TMLR — Accepted by TMLR_

### Review · Reviewer_XFPg · 2025-05-10

**Summary Of Contributions:**

This paper presents a method for approximating the behavior of a Langevin method when only a particular biased estimate of the true score function (computed via denoising score matching) is known. It is shown that a step in the proposed method gives samples that are close in distribution to those of the Langevin method using the true score function. It is tested empirically, and appears to give accurate results.

**Audience:**

Yes

**Claims And Evidence:**

Yes

**Requested Changes:**

# Critical Changes

Either strengthen the theory so that it suffices to prove the method generates approximate samples from the target distribution **or** substantially increase the amount of computational evidence that this method is doing what is intended and gives reasonable improvement over existing methods.

**Strengths And Weaknesses:**

# Strengths
- The idea itself is good.
- The writing is well-structured and reasonably clear.
- The experiments are promising.

# Weaknesses
- The theory is rather limited. It feels like a good, but incomplete, start at analysis.
    * The theorem only gives a bound for the next step when starting from a sample from the target distribution:
        - The whole point of using Langevin algorithms for sampling is to sample from a target distribution when starting from points out of the distribution.
        - Ideally, we would like to see how samples from the algorithm get closer to the target distribution
    * The error measure in terms of characteristic functions at fixed frequencies is somewhat non-standard. It would be more  compelling to compare the difference in distributions via more common measures (i.e. TV distance, Wassertein distance, KL divergence, etc.)
    * Only a single step of the algorithm is analyzed. Due to the restrictions on the starting distribution, it does not seem obvious how to extend the analysis to multiple steps (since after the first step, it will be slightly out of the target distribution.)
- The theorem statement is a bit hard to parse. In particular, an equation explicitly describing the error bound should be given. As it stands, I needed to read the appendix to even understand what error was being measured.
- In light of the limited theory, I would hope for more thorough experimental results. In particular, I would think that some comparison to the discussed related methods should be given.

---

> ### Author Response · Authors · 2025-07-30
> **Reply to reviewer comments**
>
> Thank you for your review. I don't find much to argue against on a factual basis, although of course, the amount of evidence required for publication at TMLR is subject to interpretation.
>
> I use the characteristic function since it seems natural for the analysis of an additive formula such as the Langevin iteration. It may lead to difficulties in interpretation of the bounds, that is true. In fact, Beyler and Bach ( https://arxiv.org/abs/2503.12966 ) have made a rigorous analysis of the meaning of related bounds based on the characteristic function, partly inspired by the initial version of our current manuscript.

---

### Review · Reviewer_SpXs · 2025-05-11

**Summary Of Contributions:**

The paper proposes a version of Langevin Monte-Carlo to sample from certain target distributions using only the score function of noisy data samples. The authors prove a result stating equivalence between the iterates of their algorithm and the original Langevin algorithm.

**Audience:**

No

**Claims And Evidence:**

No

**Requested Changes:**

The main theorem of the paper is not well-written and is mathematically non-informative. See the weakness section above.

**Strengths And Weaknesses:**

### Strengths

The clever design of the algorithm allows to avoid solving a denoising problem.

### Weaknesses

* There is a lower bound on the stepsize: $\mu \geq \sigma^2/2$. In general, in order for Langevin algorithm to converge, the stepsize needs to be small compared to the parameters. Thus, this implies that the noise level $\sigma^2$ has to be small. Consequently, estimating the approximate noisy score function $\Psi_{\tilde{\mathbf{x}}}$ will be (almost) as hard to compute as the original score $\Psi_{\mathbf{x}}$, hence the main goal of the proposed algorithm is not attained.
* The statement of the main theorem is not clear.

  * Why is the characteristic function considered?
  * How is the argument of the characteristic function related to the iterates of the proposed Langevin algorithm?
  * What does “one iteration of the proposed algorithm is equal to one iteration of the Oracle Langevin up to terms of order $O(\cdot)$” mean? This should be clearly stated in the main part of the paper. One should not have to read the entire proof to understand the precise meaning.
  * What happens if the norm of $\xi$ is large? Given that the characteristic function can take any argument, the norm of $\xi$ can be arbitrarily large. The similarity (let alone equality) between LMC and the proposed algorithm will not hold.
* The comparison between LMC and the proposed algorithm must be more rigorous. The theorem, in its current form, is not informative and does not provide a practically useful or interpretable guarantee.

---

> ### Author Response · Authors · 2025-07-30
> **Reply to reviewer comments**
>
> Thank you for your review.
>
> Regarding the step size, it is true that DSM cannot be used for an infinitesimal noise level; if it could, the very problem of having access only to the noisy-data score function would not occur since we could estimate the true noise-free score function up to infinitesimal perturbation. However, DSM can be used for quite small noise levels, and it is an empirical question whether DSM will allow for a noise level (and thus a step size in out algorithm) that is small enough for the our sampling to work well. This must depend a lot on the data, the model of the score function, among other things. I don't have a definite answer to that.
>
> I use the characteristic function since it seems natural for the analysis of an additive formula such as the Langevin iteration. It may lead to difficulties in interpretation of the bounds, that is true. In fact, Beyler and Bach ( https://arxiv.org/abs/2503.12966 ) have made a rigorous analysis of the meaning of related bounds based on the characteristic function, partly inspired by the initial version of our current manuscript.

---

### Review · Reviewer_2z9W · 2025-07-09

**Summary Of Contributions:**

This paper proposes a method for estimating the noisy score-matching function using the Tweedie-Miyasawa formula. Based on this, it introduces a noise-corrected Langevin algorithm aimed at reducing the bias caused by noisy score estimation (or the discretization of continuous Langevin dynamics). Morever, the paper characterizes the distributional gap between the proposed noise-corrected Langevin algorithm and the original Langevin algorithm.

**Audience:**

Yes

**Claims And Evidence:**

Yes

**Requested Changes:**

Please see the concerns in Strengths And Weaknesses.

**Strengths And Weaknesses:**

The proposed method is novel and noteworthy. Compared to conventional DSM approaches, this paper introduces a more computationally efficient method.

However, I have a few concerns outlined below:
1. Regarding the noise-corrected Langevin algorithm, it is unclear whether the bias in this paper refers to bias caused by the discretization error of Langevin dynamics, or the bias caused by the discrepancy between the score functions of the noisy data and the clean data. Clarification on this point would be helpful.

2. Using Eq. (11), it appears that the proposed noise-corrected Langevin algorithm in Eq. (6) is equivalent to running Langevin dynamics on the noisy data $\tilde{x}_t$ using Eq. (11), followed by a single corrective update using Eq. (6) at the final step. It is not clear to me how this single-step correction addresses either of the potential sources of bias mentioned above.

3. The implication of Theorem 2 is not clear to me. The theorem compares a one-step Langevin update on clean data with the proposed update in Eq. (6), which can be rewritten as:
$x_{t+1}= \tilde{x}_t + \mu \tilde{\Psi}(\tilde{x}_t) + \sqrt{2\mu - \sigma^2} \nu_t
= x_t + \sigma n_t + \mu  \tilde{\Psi}(\tilde{x}_t) + \sqrt{2\mu - \sigma^2} \nu_t
= x_t + \mu  \tilde{\Psi}(\tilde{x}_t) + \sqrt{2\mu} \nu_t$.

This suggests that the only difference between the per-step updates of the proposed method and the original Langevin algorithm lies in the score function. Given this, the error term w.r.t. $\mu$ derived in Theorem 2 seems somewhat trivial. It is unclear how this analysis implies a reduction in either discretization bias or score mismatch bias.

---

> ### Author Response · Authors · 2025-07-30
> **Reply to reviewer comments**
>
> Thank you for your thoughtful review. Here are some answers to your concerns:
>
> Point 1: I would like to clarify that the bias that is reduced here is the bias due to the discrepancy between the score functions of the clean and noisy data. The bias due to discretization is in no way reduced (as far as I can tell).
>
> Point 2: Indeed, some such interpretation of the method seems to be valid. This has actually been explored by Beyler and Bach ( https://arxiv.org/abs/2503.12966 ) , inspired by the initial version of our current manuscript. They find that for regular densities, such a half-denoising gives a reasonable sampling, while in some irregular cases, it does not. But I do not have an intuitive explanation of why such a half-denoising works, it is indeed surprising to me as well.
>
> Point 3: I believe the development the reviewer presents is not valid. First, in the last formula on the right, the noise in \tilde{x} is now dependent on the noise \nu, since the same n is now used or absorbed in both; this is against what should be the case in a valid Langevin iteration where the additive noise is independent. Second, in the same formula, the x_t appears as both the noisy and noise-free version, which is again against the definition of a Langevin iteration.

---

> > ### Comment · Reviewer_2z9W · 2025-08-06
> >
> > Thank you for your response.
> >
> > I believe the new score estimation method proposed in this paper offers meaningful value. While there is not yet conclusive evidence that it fully eliminates the bias introduced by noisy score estimation in Langevin dynamics, Theorem 2 provides a theoretical guarantee that the first-order error is removed. A comprehensive investigation would require significantly more effort and is understandably beyond the scope of this paper. Given the method's novelty and its potential for further study, I believe the current version is suitable for publication in TMLR.

---

### Decision · Action_Editor_WqBZ · 2025-08-21

**Recommendation:** Accept with minor revision

**Additional Comments:**

- The authors are encouraged to better articulate the limitations of the theorem pointed out by Reviewer XFPg.
- The authors state ''*But then there is no straightforward way of computing the $x_t$ which is not an obvious
function of $\tilde{x}_t$.*'' However, one can obtain a noise-free sample $x_t$ simply by applying the proposed iteration (6) after running the ordinary LMC (11) for the noisy-data distribution. In other words, the proposed method is essentially applied only at the final iteration of an ordinary LMC, which may give readers the impression that its effect is limited. It would be helpful if the authors could clarify this point in the paper.

**Audience:**

Yes

**Audience Explanation:**

This paper could be of interest to researchers in the field of sampling and generation.

**Claims And Evidence:**

Yes

**Claims Explanation:**

This paper proposes a noise-corrected Langevin algorithm that leverages a noisy-data score function, inspired by LMC and denoising score matching (Tweedie-Miyasawa formula). While the noisy-data score introduces bias compared to the oracle LMC, the authors reduce this bias by lowering the Gaussian noise weight. Theorem 2 provides a nontrivial theoretical guarantee, supported by a characteristic-function-based argument. Reviewers found the idea interesting and potentially impactful, but noted the results are limited in scope: the analysis does not establish convergence, leaving the theoretical picture incomplete. Nonetheless, the bias-reduction guarantee holds in the sense of Theorem 2, and the direction has the potential to inspire further work.